# Dehydroabietic Acid Microencapsulation Potential as Biofilm-Mediated Infections Treatment

**DOI:** 10.3390/pharmaceutics13060825

**Published:** 2021-06-02

**Authors:** Iris Neto, Eva María Domínguez-Martín, Epole Ntungwe, Catarina P. Reis, Milica Pesic, Célia Faustino, Patrícia Rijo

**Affiliations:** 1CBIOS-Research Center for Biosciences & Health Technologies, Universidade Lusófona de Humanidades e Tecnologias, Campo Grande 376, 1749-024 Lisboa, Portugal; irisrbneto@gmail.com (I.N.); evam.dominguez@uah.es (E.M.D.-M.); p5999@ulusofona.pt (E.N.); 2Instituto de Investigação do Medicamento (iMed.ULisboa), Faculdade de Farmácia, Universidade de Lisboa, 1649-003 Lisboa, Portugal; catarinareis@ff.ulisboa.pt; 3Pharmacology Area (Pharmacognosy Laboratory), New Antitumor Compounds: Toxic Action on Leukemia Cells Research Group. Ctra. A2, Department of Biomedical Sciences, Faculty of Pharmacy, Km 33.100—Campus Universitario, University of Alcalá de Henares, Alcalá de Henares, 28805 Madrid, Spain; 4Institute for Biological Research “Sinisa Stankovic”-National Institute of Republic of Serbia, University of Belgrade 142, 11060 Belgrade, Serbia; camala@ibiss.bg.ac.rs

**Keywords:** dehydroabietic acid, antimicrobial resistance, biofilm, infection, microencapsulation

## Abstract

The antimicrobial activity of dehydroabietic acid (DHA) for its use as an antibiofilm agent was tested in this work. DHA was assayed against a collection of Gram-positive, Gram-negative sensitive and resistant bacteria and yeasts through the minimum inhibitory concentration (MIC), MIC with Bioburden challenge, minimum bactericidal concentration (MBC), minimum biofilm inhibitory concentration (MBIC), MBIC with Bioburden challenge and growth curve studies. Toxicological studies (*Artemia salina*, sulforhodamine B (SRB) assay) were done to assess if the compound had antimicrobial and not cytotoxic properties. Furthermore, microencapsulation and stability studies were carried out to evaluate the chemical behavior and stability of DHA. On MIC results, Gram-positive bacteria *Staphylococcus aureus* ATCC 1228 and *Mycobacterium smegmatis* ATCC 607 presented a high efficiency (7.81 µg/mL), while on Gram-negative bacteria the highest MIC value of 125 µg/mL was obtained by all *Klebsiella pneumoniae* strains and *Escherichia coli* isolate strain HSM 303. Bioburden challenge showed that MIC, MBIC and percentage biofilm inhibition (BI) values suffered alterations, therefore, having higher concentrations. MBIC values demonstrated that DHA has a higher efficiency against *S. aureus* ATCC 43866 with a percentage of BI of 75.13 ± 0.82% at 0.49 µg/mL. Growth curve kinetic profiles of DHA against *S. aureus* ATCC 25923 were observed to be bacteriostatic. DHA-alginate beads had a average size of 2.37 ± 0.20 and 2.31 ± 0.17 × 10^3^ µm^2^ with an encapsulation efficiency (EE%) around 99.49 ± 0.05%, a protection percentage (PP%) of 60.00 ± 0.05% in the gastric environment and a protection efficiency (PE%) around 88.12 ± 0.05% against UV light. In toxicological studies DHA has shown IC_50_ of 19.59 ± 7.40 µg/mL and a LC_50_ of 21.71 ± 2.18%. The obtained results indicate that DHA is a promising antimicrobial candidate against a wide range of bacteria and biofilm formation that must be further explored.

## 1. Introduction

Antimicrobial resistance (AMR) is one of the current worldwide problems in developing and developed countries, because of the misuse and overuse of antibiotics [1]. In fact, among the main causes of morbidity and mortality in developing countries are infectious diseases, causing the healthcare-acquired infections (HAI) high rates of deaths globally [2]. Actually, it is estimated that there are 700,000 deaths worldwide related to AMR, and that number will rise in 2050 approximately to 10 million deaths [3].

Currently, some microorganisms’ strains have become common inhabitants in the hospital environments, such as methicillin-resistant *Staphylococcus aureus* (MRSA), vancomycin-resistant *S. aureus* (VRSA), vancomycin-resistant *Enterococcus* (VRE) and multidrug-resistant (MDR) [4,5,6]. The problem aggravates considering the diminution of new antibiotics development, raising the infections specter that once were treatable may soon become untreatable [7]. This justifies why the World Health Organization (WHO) identified antimicrobial resistance as a public health concern just at the beginning of the 21st century [8], as a consequence of the quickly arise and spread of resistant microorganisms to commonly used antibiotics.

AMR to regularly employed antimicrobial agents is primarily a result of the enzymatic inactivation of the drug, a shift of its receptor or target site, altered membrane permeability (which prevents the drug to access to the inner of the bacteria) or the overexpression of efflux pumps [9,10,11]. Moreover, in the past, many antibiotics were developed considering the microorganisms grew in planktonic cultures, but currently it is clear that most of the bacteria live as complex communities known as biofilms. Undoubtedly, as current published works evidenced, microbial biofilms are the responsible for the treatment failure to conventional antimicrobial therapy, due to the difficulty of these drugs to penetrate and destroy biofilm matrix [12,13,14]. Therefore, to contain and to solve the global increasing of AMR, it is necessary to invest on novel classes of antimicrobial agents with better efficacy and mechanisms of action, to be used alone or as combination regimens with another antibiotics. Isolated compounds from herbal medicines can enrich our AMR therapeutic arsenal used solely or in combination with the current antibiotics available in the market.

Our research group has recently isolated several antimicrobial diterpenes from *Plectranthus* plants, namely royleanones and coleons, which belong to the largest group of naturally occurring abietanes [15,16,17]. The abietanes possess a characteristic aromatic ring C, being an example, the dehydroabietic acid (DHA) (Figure 1). The abietic acids are strong antibacterial agents with DHA generally being the most potent [18].

Antimicrobial diterpenes like DHA target cellular membranes by a combination of hydrophobic and electrostatic adsorption effects at the membrane/water interface, leading to membrane destabilization and enhanced permeability. Subsequently, disruption of the physical integrity of the membrane or translocation into the cell occurs, compromising cellular processes such as DNA replication, protein folding and synthesis. This mode of action has been shown to limit the risk of cross-resistance [19,20,21].

The present work is focused on the antimicrobial and toxicological studies of DHA, and DHA-alginate beads were produced for protection of DHA from degradation.

## 2. Materials and Methods

### 2.1. Reagents

Phosphate buffered saline (PBS), alginate, sulfuric acid, absolute ethanol (99.7%), trichloroacetic acid, sulforhodamine B, acetic acid and tris(hydroxymethyl)aminomethane (TRIS base) were purchased from Sigma-Aldrich^®^ (Darmstadt, Germany). Gram’s crystal violet solution, dimethyl sulfoxide (DMSO), deuterated dimethyl sulfoxide (DMSO-d6), anhydrous chloroform, deuterated chloroform (CDCl_3_), methanol, ethyl acetate (AcOEt) and calcium chloride were bought from Merck^®^ (Lisboa, Portugal). Dehydroabietic acid was obtained from Wako^®^ (Neuss, Germany) (FUJIFILM Wako Pure Chemical Corporation; CAS RN^®^: 1740-19-8, Molecular Formula: C20H28O2; Molecular Weight: 300.44). Mueller–Hinton Broth (MHB), Mueller–Hinton agar (MHA), Sabouraud dextrose broth (SDB), Sabouraud dextrose agar (SDA) and tryptic soy broth (TSB) were purchased from Biokar^®^ Diagnostics (Allonne, France).

### 2.2. Equipment

The microbiological assays required Greiner^®^ Bio-one 96 well V-bottom microplates, Costar^®^ cell culture 96 well flat-bottom microplates, a microplate reader from Thermo-Fisher Scientific (Marsiling, Singapore), incubators from Memmert, Heidolph Inkubator 1000 and Heidolph Unimax 1010 Stirrer (Heidolph Instruments GmbH & Co. KG, Schwabach, Germany), a laminar flow cabinet from Faster Bio 48 (Milan, Italy), Xiaomi Redmi 4 (5-element lens, 1.12 μm pixels, 13 MP CMOS) Camara and a spectrophotometer from Hitachi (Santa Clara, CA, USA).

### 2.3. Dehydroabietic Acid Purification and Identification

DHA obtained from Wako^®^ was purified through the process of recrystallization from methanol. The solid crystals of DHA were collected and the purity was checked by nuclear magnetic resonance (NMR) spectroscopy. The chemical structure of purified DHA was elucidated based on 1D-(^1^H and ^13^C) and 2D (COSY, HSQC and HMBC) NMR spectroscopy experiments. NMR spectra were recorded in DMSO-d6 and CDCl_3_ on a 400 MHz instrument (Varian INOVA-400 Spectrometer (^UNITY^INOVA™ NMR Spectrometer Systems) at 400 MHz (^1^H) and 75 MHz (^13^C). The chemical shifts (δ) are in ppm and the coupling constants (*J*) in Hertz (Hz).

### 2.4. Bacterial Strains and Culture Media

Preservation and culture of the test organisms were carried out accordingly to the Clinical and Laboratory Standards Institute (CLSI). Bacterial strains studied were Gram-positive bacteria (*Staphylococcus aureus* ATCC 25923 (MSSA), *S. aureus* CIP 106760 (MRSA), *S. aureus* ATCC 6538, *S. aureus* ATCC 25923, *S. aureus* ATCC 700699, *S. aureus* FFHB 25923 (MRSA), *S. aureus* 43300, *S. aureus* ATCC 43866, *S. epidermidis* ATCC 12228, *Enterococcus faecalis* ATCC 29212, *E. faecalis* ATCC 51299, *E. faecalis* V583 and *Mycobacterium smegmatis* ATCC 607), Gram-negative bacteria (*Pseudomonas aeruginosa* ATCC 9027, *Escherichia coli* ATCC 25922, *E. coli* HSM 1323, *E. coli* HSM 1465, *E. coli* HSM 3023, *Klebsiella pneumoniae* HSM 701, *K. pneumoniae* HSM 703, *K. pneumoniae* HSM 873, *K. pneumoniae* HSM 2948 and *Enterobacter cloacae* HSM 1284), a diploid fungus (*Candida albicans* ATCC 10231) and yeast (*Saccharomyces cerevisiae* ATCC 2601). For the growth and maintenance of the microorganisms, MHB, MHA, SDB and SDA were used, whereas for creating the bacterial biofilm, TSB supplemented by 1% glucose was employed. All microorganisms were cultured and incubated during 24 h at 37 °C in an aerobic workstation. In order to harvest overnight cultures, bacterial suspensions used for antibacterial assays and quantitative biofilm assays were prepared, suspending them in sterile water and adjusting the turbidity to 0.5 McFarland standard.

### 2.5. Bioburden Challenge

Two different organic challenges were investigated, namely 4% bovine serum albumin (BSA, Sigma-Aldrich^®^, Darmstadt, Germany) and human blood serum (HBS, Sigma-Aldrich^®^).

### 2.6. Cell Cultures

HaCaT cell line (normal human keratinocytes obtained from CLS-Cell Lines Service, Eppelheim, Germany). In a Dulbecco’s modified eagle’s medium (DMEM) supplemented with 10% fetal bovine serum (FBS), 4 g/L glucose, l-glutamine (2 mM) and 5000 U/mL penicillin, 5 mg/mL streptomycin solution and HaCaT cells were cultured. Then, HaCaT cells were subcultured at 144 h intervals using 0.25% tryp-sin/ethylenediaminetetraacetic Acid (EDTA) and seeded into a fresh medium at 64,000 cells/cm^2^.

### 2.7. Antimicrobial Tests

#### 2.7.1. Minimum Inhibitory Concentration Assay

The minimum inhibitory concentration (MIC) of DHA was determined through the two-fold serial broth microdilution assay following CLSI guidelines [22]. In aseptic conditions, 100 μL of the medium (MHB or SDB) was distributed in each well of a 96-well plate. Later, to the first well of each row was added 100 μL of DHA (diluted in DMSO), positive control (commercial antimicrobial: rifampicine (RIF) and vancomycin (VAN) for Gram positive bacteria; ampicillin (AMP) for Gram negative bacteria; nystatin (NYS) for yeasts;), or negative control (sterile culture media) solutions at 1 mg/mL concentration. Additionally, DMSO was used as negative control as the solvent used to dissolve DHA, at this high concentration DMSO was not toxic as previously described [15,16,17,23]. Using a multichannel micropipette, a serial dilution was made to 1:2 proportion between each row of wells (50,000–0.49 mg/L range). Finally, 10 μL of bacterial suspension was added to every well, and plates were covered and incubated at 37 °C for 24 h. The bacterial growth was measured with an absorbance microplate reader (Thermo Scientific Multiskan FC, Loughborough, UK) set to 595 nm. Assays were carried out at least in triplicate for each tested microorganism.

#### 2.7.2. Minimum Bactericidal Concentration Assay

The minimum bactericidal concentration (MBC) was determined according to the procedure recommended by the CLSI [22]. MBC was defined as the lowest concentration of DHA at which a complete absence of growth of bacterial colonies on the agar surface was observed compared with a non-treated control. Assays were carried out at least in triplicate for each tested microorganism.

#### 2.7.3. Minimum Inhibitory Concentration Assay with the Bioburden Challenge

The MICs with the Bioburden challenge were measured as described above (Section 2.7.1). After the addition of the bacterial suspension to the wells, 10 µL of BSA 4% and HBS solutions were added to the wells, and the plates were incubated for 24 h at 37 °C. The bacterial growth was measured with an absorbance microplate reader set to 595 nm. Assays were carried out at least in triplicate for each tested microorganism.

#### 2.7.4. Biofilm Formation Assay

Biofilm formation of the selected strains was detected by an optimized semiquantitative assay in 96-well polystyrene microtiter plates as previously described [23]. Overnight cultures of clinical strains were diluted at 0.5 McFarland standard in TSB containing 1% glucose and dispensed into 96-well flat-bottom plates (300 µL/well). After 24 h of static incubation at 37 °C, the plates were washed gently three times with PBS (pH 7.2) to remove unattached bacteria, fixed at 60 °C for 60 min in an incubator, and stained with crystal violet 1% (*w*/*v*) for 15 min at room temperature. Wells were then thoroughly washed with running tap water. Biofilm formation was quantified by adding 150 µL of absolute ethanol into each well for 30 min, and then scanned at 595 nm using a 96-well plate spectrophotometer to determine the optical density of the stained biofilms. Sterile culture media was used as a negative control.

According to their biofilm-forming ability, the selected strains were divided into four groups, as measured by OD values at 595 nm (OD_595_). The cut-off OD value (ODc) was defined as three standard deviations (SD) above the mean OD of the negative control (Equation (1)). The OD value of a tested strain was expressed as its average. The strains were divided into the following groups: OD_595_ ≤ ODc = no biofilm producer (-); ODc < OD_595_ ≤ 2 × ODc = weak biofilm producer (+); 2 × ODc < OD_595_ ≤ 4 × ODc = moderate biofilm producer (++); 4 × ODc < OD_595_ = strong biofilm producer (+++). Assays were carried out at least in triplicate for each tested microorganism [24].
(1)$${\mathrm{OD}}_{c}=\overset{¯}{\underset{\;}{X}}\times \mathrm{OD}\times \mathrm{Negative}\;\mathrm{control}+(3\times \mathrm{SD}\times \mathrm{Negative}\;\mathrm{control})$$

#### 2.7.5. Inhibition Properties of Dehydroabietic Acid on Biofilm Formation

By two-fold dilution and crystal violet staining methods, the effect of DHA on biofilm formation was assayed.

The minimum biofilm inhibitory concentration (MBIC) of the compounds was determined using the two-fold serial broth microdilution assay. In aseptic conditions, 100 μL of TSB containing 1% glucose medium was distributed in each well of a 96-well microplate. Next, to the first well of each row were added 100 μL of DHA, the positive control or negative control solutions at 1 mg/mL concentration. Using a multichannel micropipette, a serial dilution was made to 1:2 proportion between each row of wells (50,000–0.49 mg/L range). Lastly, 20 μL of bacterial suspension was added to every well, and plates were covered and incubated at 37 °C for 24 h. After the crystal violet staining method, the biofilm growth was measured with an absorbance microplate reader set to 595 nm. The percentage of biofilm formation in the presence of different concentrations of DHA was determined using Equation (2). Results are shown as mean ±SD, *p* < 0.05.
(2)$$\mathrm{Biofilm}\;\mathrm{Formation}\;\left(\%\right)=\frac{{\mathrm{OD}}_{595}\mathrm{of}\;\mathrm{the}\;\mathrm{test}\;\mathrm{well}}{{\mathrm{OD}}_{595}\;\mathrm{of}\;\mathrm{non}-\mathrm{treated}\;\mathrm{control}\;\mathrm{well}}\times 100$$

#### 2.7.6. Inhibition Properties of Dehydroabietic Acid on Biofilm Formation with Bioburden Challenge

The effect of DHA on biofilm formation with interfering substances was determined by two-fold microdilution and crystal violet staining methods, described previously (Section 2.7.1 and Section 2.7.4, respectively) [23,24]. The MBIC was determined employing the two-fold serial broth microdilution assay. A total of 100 μL of TSB containing the 1% glucose medium were introduced in each well of a 96-well plate under aseptic conditions. Next, to the first well of each row 100 μL of DHA was added, the positive control or negative control solutions at a 1 mg/mL concentration. Using a multichannel micropipette, a serial dilution was made to 1:2 proportion between each row of wells (50,000–0.49 mg/L range). Finally, 10 μL of bacterial suspension plus 10 μL of BSA 4% or HBS solutions were added to every well, and plates were covered and incubated at 37 °C for 24 h.

After the crystal violet staining method, the biofilm growth was measured with an absorbance microplate reader set to 595 nm. Using Equation (2), the percentage of biofilm formation in the presence of different concentrations of each compound was established. Assays were carried out at least in triplicate for each tested microorganism. Results are shown as mean ± SD, *p* < 0.05.

#### 2.7.7. Growth Curve Studies

The growth curve of DHA was determined using the method described previously [25]. Concentrations equivalent to the MIC of DHA were prepared. After overnight incubation over stirring (at 100 RPM), inoculum size of 1.0 × 10^5^ colony forming units (CFU)/mL was chosen. DHA was prepared at different concentrations that were incorporated into test tubes containing sterile nutrient broth and inoculated with test microorganisms. Aliquots of 1.0 mL of the medium taken at time intervals of 30 min for 6 h were inoculated into nutrient agar and incubated at 37 °C for 24 h and measured with a spectrophotometer set to 620 nm. A control test (negative) consisted of test microorganisms alone without DHA.

After 24 h of incubation, the CFU was determined (data not shown). The procedure was repeated in triplicates (three independent experiments) and a graph of the absorbances against time was plotted.

### 2.8. Microencapsulation of Dehydroabietic Acid

#### 2.8.1. Production of Dehydroabietic Acid-Loaded Calcium Alginate Microspheres

By the extrusion/external gelation method, DHA-alginate microspheres were prepared [26]. To a 5 mL of 2% sodium alginate solution, a volume of 1 mL of reconstituted DHA at 10 mg/mL was incorporated. Under magnetic stirring, this suspension was extruded through a fine syringe (21-gauge, diameter 0.81 mm) to 100 mL of calcium chloride 0.05 M solution for 15 min.

Gelled and hydrated microspheres were recovered by filtration. Blank microspheres were also produced.

#### 2.8.2. Characterization of Dehydroabietic Acid-Loaded Calcium Alginate Microspheres

The size and morphology of the calcium alginate beads were determined by direct observation, and the images were recorded digitally (Xiaomi Redmi 4 Camara). The encapsulation efficiency (EE%) of DHA-alginate microspheres was determined in triplicate through an indirect method. The absorbance (Abs_620nm_) of the supernatant obtained in the preparation of the microspheres was measured through a spectrophotometric method previously described [27,28]. In Equation (3), as an indicator of the EE% were used calcium chloride and DHA concentration, where EE% corresponds to the encapsulation efficiency, C_i_ to the initial concentration of DHA and C_f_ to the final concentration of DHA (non-encapsulated or free DHA) present in the supernatant. Results are shown as mean ± SD, *p* < 0.05.
(3)$$\mathrm{Encapsulation}\;\mathrm{efficiency}\;\left(\%\right)=\frac{C_{i}-C_{f}}{C_{i}}\times 100$$

#### 2.8.3. Stability Studies of Dehydroabietic Acid-Loaded Calcium Alginate Microspheres

##### Harsh Conditions Such as Gastric Environment or Low pH Study

To test the protection percentage (PP%) for DHA-alginate microspheres in acidic conditions, a forced degradation study was conducted. It was performed the acid stress reflux with HCl (0.1 M) at 60 °C for 30 min as beforehand defined [29,30,31]. With the same spectrophotometric method previously described the amount of DHA in the supernatant and remaining DHA in beads were quantified.

Fresh solutions of DHA and DHA-alginate microspheres were used as the controls. DHA concentration was used as the indicator of PP% using Equation (4), where PP% corresponds to the protection efficiency, C_i_ to the initial concentration of DHA and C_f_ to the final concentration of DHA (non-encapsulated or free DHA) present in the supernatant. Results are shown as mean ± SD, *p* < 0.05.
(4)$$\mathrm{Protection}\;\mathrm{percentage}\;\left(\%\right)=\frac{C_{i}-C_{f}}{C_{i}}\times 100$$

##### Photostability Studies

Freeze-dried beads made from calcium alginate (200 mg) were deposited in vials containing 10 mL of water under UV light (274 nm) and left for magnetic stirring for 2 h, as previously described [26].

The amount of DHA in the supernatant and remaining in beads was quantified using the same spectrophotometric method. Fresh solutions of DHA and DHA-alginate microspheres were used as the controls. Calcium chloride and DHA concentration were used as the indicator of protection efficiency using Equation (5), where PE% corresponds to the protection efficiency, C_i_ to the initial concentration of DHA and C_f_ to the final concentration of DHA (non-encapsulated or free DHA) present in the supernatant. Results are shown as mean ±SD, *p* < 0.05.
(5)$$\mathrm{Protection}\;\mathrm{efficiency}\;\left(\%\right)=\frac{C_{i}-C_{f}}{\mathrm{Ci}}\times 100$$

### 2.9. Toxicological Studies

#### 2.9.1. Sulforhodamine B Assay

HaCaT cells grown in 25 cm^2^ tissue flasks were trypsinized, seeded into flat-bottom 96-well tissue culture plates, and incubated overnight. Later, they were seeded at 4000, 8000 and 16,000 cells/well, respectively.

Compound used as treatment were incorporated at concentration of 1–100 mM lasted 72 h. The cellular proteins were stained with sulforhodamine B (SRB) assay, following the slightly modified protocol previously described [32]. Briefly, the cells in 96-well plates were fixed in 50% trichloroacetic acid (50 mL/well) for 1 h at 4 °C, rinsed in tap water and stained with 0.4% (*w*/*p*) sulforhodamine B in 1% acetic acid (50 mL/well) for 30 min at room temperature. The cells were then rinsed three times in 1% acetic acid to remove the unbound stain. The protein-bound stain was extracted with 200 mL 10 mM TRIS base (pH 10.5) per well. The optical density was read at 540 nm, with correction at 670 nm. Results are shown as mean ±SD, *p* < 0.05.

#### 2.9.2. Artemia Salina Toxicological Study

The toxicity of DHA on *Artemia salina* was tested according to the method described previously [33,34,35], with some small adaptations on the hatching equipment, namely material for covering the compartments. DHA concentrations of 10 ppm were tested.

In artificial seawater (salinity concentration 30 g/L), during 48 h at 30 °C, 80 mg of brine shrimp cysts (JBL GmbH & Co., KG D-67141; Neuhofen, Germany) were incubated. Next, from the hatched cysts, ten nauplii were selected and transferred into wells of 21-well cultures plates containing artificial seawater (final volume/well 1 mL). The culture plates were incubated for 48 h at 30 °C; after every 24 h, the number of dead nauplii was counted microscopically.

The mortality was used to calculate the percentage (%) of lethal concentration, according to Equation (6). Results are shown as mean ± SD, *p* < 0.05.
(6)$$\mathrm{Lethal}\;\mathrm{concentration}\;\left(\%\right)=\frac{\mathrm{Total}\;\mathrm{nauplii}-\mathrm{Alive}\;\mathrm{nauplii}}{\mathrm{Total}\;\mathrm{nauplii}}$$

### 2.10. Statistical Analysis, Software and Databases

In order to perform statistical analysis, GraphPad Prism version 7.0a (San Diego, CA, USA,) and a one-way ANOVA were used with a significance level (*p*-value) of 0.05. Additionally, the following software and database were used: ChemDraw 19.0 (PerkinElmer Informatics, Massachusetts, USA), MestreNova 8.1.4 (York, England), Microsoft Excel (Microsoft Office 365), PubMed-NCBI and ISI web of knowledge.

## 3. Results

### 3.1. Dehydroabietic Acid Recrystallization

Commercial DHA (purity 85–90%) was recrystallized from methanol to give white needle crystals and purity was verified by NMR. ^1^H NMR(DMSO-d6, 400 MHz, ppm) δ 7.08 (d, 1H, *J_11,12_* = 8.1 Hz, H-11), 6.89 (d, *J_12,11_* = 8.1 Hz, 1H, H-12), 6.77 (brs, 1H, H-14), 2.72-2.68 (m, 2H, H-7, H-15, overlapped signals), 2.40-2.45 (m, 4H, 4H, H-2α, H-2β, H-3α, H-3β), 2.22 (brs, 1H, H-5), 1.90-1.98 (m, 2H, H-6α and H-6β), 1.58 (m, 1H, H-1α), 1.25 (m, 1H, H-6β), 1.08 (brs, 6H, Me-16, Me-17, overlapped signals), 1.07 (s, 3H, Me-19), 1.04 (brs, 3H, Me-20); ^13^C NMR (DMSO-d6, 75 MHz, ppm) δ 179.53 (C-18), 146.80 (C-9), 145.11 (C-13), 134.16 (C-8), 126.53 (C-14), 124.12 (C-11), 123.79 (C-12), 46,43 (C-5), 44.76 (C-4), 37.83 (C-1), 36.49 (C-3), 36.32 (C-10), 32.95 (C-15), 29.61^a^ (C-17), 24.89 (C-19), 24.01 (C-20), 21.19^a^ (C-16), 18.23 (C-6) and 16.44 (C-2). ^a^ May be interchanged.

### 3.2. Antimicrobial Tests

#### 3.2.1. Minimum Inhibitory Concentration Assay

The antimicrobial activity of DHA was screened and MIC values were determined against a collection of Gram-positive and Gram-negative bacteria and yeasts using the microdilution method. The results obtained are shown in Table 1.

In Gram-positive reference bacteria, *S. epidermidis* ATCC 12228 and *M. smegmatis* ATCC 607, DHA demonstrated a high efficiency with a MIC value of 7.81 µg/mL (Table 1). Regarding bacteria *S. aureus* CIP 106760, a MIC value of 15.63 µg/mL was showed by DHA, having even higher efficiency than the positive control. Against the remaining bacterial pathogens tested, DHA also had a potential antibacterial effect.

Regarding Gram-negative microorganisms (Table 1), the lowest MIC value (125 µg/mL) was registered against *K. pneumoniae* isolate strains and *E. coli* isolate strain HSM 3023.

#### 3.2.2. Minimum Bactericidal Concentration

Bactericidal activity of DHA was also screened against a collection of Gram-positive (MSSA *S. aureus* ATCC 25923 and *S. epidermidis* ATCC 12228), Gram-negative bacteria (*E. coli* ATCC 25922 and slime producer *P. aeruginosa* ATCC 9027) and yeasts (*C. albicans* ATCC 10231 and *S. cerevisiae* ATCC 2601) using the microdilution method. Regarding MBC values, low MBC/MIC ratios within each strain are typical of bactericidal agents. In a recent study [36], the abietane-type diterpenoid taxodone showed MIC and MBC values of 500 and 500 µg/mL, respectively, against both *S. aureus* and *E. coli*. However, MBC values against the specified microorganisms could not be determined for DHA in the tested concentration range, contrary to MIC values, suggesting that antimicrobial properties of DHA are mainly bacteriostatic in nature.

#### 3.2.3. Minimum Inhibitory Concentration Assay with Bioburden

Antimicrobial activity of DHA with bioburden was screened and its MIC values were determined against Gram-positive (*S. aureus* ATCC 25923 and *S. aureus* CIP 106760) and Gram-negative (*P. aeruginosa* ATCC 9027) bacteria using the microdilution method. These strains were chosen due to their high resistance to antimicrobials (MRSA species), biofilm formation and slime production (*P. aeruginosa* ATCC 9027).

After data analysis, MIC values with bioburden suffer alterations, thus, having higher concentrations (Table 2).

#### 3.2.4. Inhibition Properties of Dehydroabietic Acid on Biofilm Formation

Biofilm formation of a collection of Gram-Positive bacteria, Gram-negative bacteria and yeasts was tested by a microdilution method and the results are shown on Table 3. Among all the strains tested only clinical isolate *E. cloacae* HSM 1264 did not produce biofilm.

The strains that are strong biofilm producers are *S. aureus* ATCC 25923, *S. aureus* ATCC 43866, clinical isolate *E. coli* HSM 3023, clinical isolate *K. pneumoniae* HSM 873 and clinical isolate *K. pneumoniae* HSM 2948. None of the Gram-negative strains demonstrated to be a moderate biofilm producer, nevertheless, in Gram-positive bacteria the strains that showed this capacity were *S. aureus* ATCC 6538, *S aureus* ATCC 700699, *S. aureus* ATCC 43300, *S. epidermidis* ATCC 12228 and all the *E. faecalis* strains. In Gram-positive bacteria only *S. aureus* CIP 106760 demonstrated to be a weak biofilm producer followed by Gram-negative clinical isolates *E. coli* HSM 1323, *E. coli* HSM *1465*, *E. coli* ATCC 25922 *K. pneumoniae* HSM 701, *K. pneumoniae* HSM 703, *P. aeruginosa* ATCC 9027 and by the yeast *C. albicans* ATCC 10231.

The MBIC of DHA on biofilm formation against a collection of Gram-positive and Gram-negative bacteria and yeasts was tested by means of the two-fold serial broth microdilution assay and results are also presented on Table 3.

The lowest MBIC value registered for DHA was achieved against Gram-positive *S. aureus* ATCC 43866, a strong biofilm producer, with a percentage of biofilm inhibition (BI) of 75.13 ± 8.82%, being achieved at 0.49 µg/mL DHA, followed by clinical isolate *K. pneumoniae* HSM 701 and clinical isolate *K pneumoniae* HSM 703. Still according to Table 3, in Gram-positive microorganisms the lowest MBIC values were in the following order: *S. aureus* FFHB 25923 < *E. faecalis* ATCC 51229 < *E. faecalis* ATCC 29212 < *E. faecalis* V583 < *S. aureus* CIP 160760 < *S. aureus* ATCC 25923 < *S. aureus* ATCC 43300 < *S. aureus* ATCC 6538 < *S. aureus* ATCC 700699. In Gram-negative bacteria, the lowest values of MBIC were attained against clinical isolates *K. pneumoniae HSM 701* and *K. pneumoniae HSM 703* followed by clinical isolate *E. coli* HSM 1465, *P. aeruginosa* ATCC 9027, clinical isolates *K. pneumoniae* HSM 873, *K. pneumoniae* and *E. coli* HSM 3023 and finally *E. coli* ATCC 25922.

Although MBIC concentrations varies between the bacterial strains, BI is never below 50% except for *S. aureus* ATCC 6538 and clinical isolate *E. coli* HSM 1323, which indicates that DHA has a strong biofilm inhibition ability that merits further investigation. Therefore, inhibition properties of DHA on biofilm formation with Bioburden and growth curve studies were performed as described in the next sections.

#### 3.2.5. Inhibition Properties of Dehydroabietic Acid on Biofilm Formation with Bioburden

In developed countries, a high economical hurdle is the high cost of chronic wounds, which are characterized by microbial complications such as local or overt infection, delaying in the healing time and spread of multiresistant microorganisms [37]. In a hope of treating germs in wounds, antisepsis is the method of choice, considering that systemic antibiotics can barely penetrate into wound biofilms and topically applied ones can easily lead to sensitization, antisepsis is the method of choice to treat germs in wounds [37].

Therefore, a MBIC study with Bioburden was conducted to demonstrate that DHA MBIC and BI do not suffer significant alterations in the presence of Bioburden. MBIC values with Bioburden for DHA was tested by means of the two-fold serial broth microdilution assay and the results are shown on Table 4.

In *S. aureus* strains, BSA demonstrates not to have any influence in MBICs, although it has in BIs values. Nevertheless, HBS has influence in both MBICs and BIs values, increasing them. In Gram-negative bacteria (*P. aeruginosa* ATCC 9027) both BSA and HBS have influence in both MBICs and BIs values by increasing them. This increase may also be due to the production of slime that this strain is known for.

#### 3.2.6. Growth Curve Study

To study the pharmacodynamics of antimicrobial agents, the growth curve assay is the method of choice, since it examines the rate at which different concentrations of an antimicrobial drug kill bacteria, displaying the concentration-dependent and time-dependent bactericidal activities of antimicrobial agents [38].

Growth curve profiles of DHA against Gram-positive bacteria (*S. aureus* ATCC 25923) were observed to be bacteriostatic. The growth curve profile of DHA against the test microorganisms at test concentrations showed reduction in number of cells over the first four hours, when compared to the control (microorganism without antimicrobial agent). The area under the curve, which is a measure of the activity of DHA over the 15 h period, showed that DHA at concentrations of 31.25, 62.50 and 93.75 µg/mL significantly (*p* < 0.05) reduced the number of *S. aureus* ATCC 25923 cells when compared to the control (Figure 2).

### 3.3. Microencapsulation of Dehydroabietic Acid

#### 3.3.1. Production and Characterization of Dehydroabietic Acid-Loaded Calcium Alginate Microspheres

Using the extrusion/external gelation method, DHA-alginate microspheres were successfully prepared. This technique is promptly performed, and it can be easily adapted to the industrial scale. Essentially, there are two main techniques for producing calcium alginate microspheres ((1) external and (2) internal gelation), based on the source of calcium. In (1), the external gelation method, the drops of alginate (name as droplets are gelled in a calcium chloride solution, and therefore, the calcium source is external to the droplet. In this case, gelation is initiated from the surface and works toward the inner core. On the other hand, in (2) the internal gelation method uses an inner source of calcium, where the alginate solution is preloaded within soluble calcium salt, releasing calcium ions with pH adjustment [26].

In the present work, DHA-alginate microspheres were spherical in shape and showed a mean diameter of 2.37 ± 0.20 and 2.31 ± 0.17 × 10^3^ µm for loaded and empty particles, respectively, in the digital images (Figure 3). These images confirmed the encapsulation of DHA inside the calcium alginate microspheres since empty calcium alginate beads were transparent; meanwhile, when DHA was present, the beads changed to yellow.

As a polyelectrolyte hydrocolloid, alginate has the property to interact with water, reducing its diffusion and stabilizing its presence. Such water may be retained specifically through direct hydrogen bonds or the structuring of water within extensive, but contained, inter- and intramolecular voids [26].

Therefore, the diterpene DHA was successfully encapsulated into calcium alginate microspheres, where the EE% value was very high (around 99.49 ± 0.05%). Alginate gelling features strongly depend upon its monomeric composition, sequential arrangements and the lengths of the G-blocks [26]. As previously mentioned in this work, alginate has the ability to form gels with calcium, and this is an autocooperative reaction that happens between calcium and the homopolymeric blocks of G on the alginate chain [26]. Calcium alginate gels shrink during gel formation, which leads to the loss of water and a rise in the concentration of polymer in the beads relative to that in the alginate solution. That may diminish some of the encapsulate diffusion to outside of the polymeric matrix. Herein, the physical characteristics of calcium alginate gel are also influenced by the alginate concentration and molecular size, the calcium concentration and gelling time [26]. Moreover, in the extrusion/external gelation method, a solution of sodium alginate is forced into a bath of a calcium chloride solution, and then, calcium alginate is formed as fibers (gel). Whether alginates of low viscosity are employed, a strong solution is normally created without any viscosity problems, and the calcium in the bath is not diluted as rapidly. Regularly, low G alginates, as the one used in this study, generate very flexible gels [26]. Gel fibers have very good strength in both wet and dry forms, and as with majority of polymer fibers made by extrusion, stretching while forming arises the linearity of the polymer chains and the strength of the fiber [26].

#### 3.3.2. Stability Studies of Dehydroabietic Acid-Loaded Calcium Alginate Microspheres

##### Harsh Conditions Such as Gastric Environment or Low pH Study

Hydrolysis, a chemical process that consists of the decomposition of a chemical compound with water, is one of the most common degradation chemical reactions over a wide range of pH [29,30,31]. Hydrolytic study under acidic and basic conditions involves catalysis by ionizable functional groups present in the molecule. For example, HCl and NaOH are employed for generating acidic and basic stress samples, respectively. The hydrolytic degradation of a new drug or molecule in acidic and alkaline condition can be studied by refluxing the drug in 0.1 M HCl/0.1 M NaOH [29,30,31]. When a reasonable degradation is detected, the test can be stopped at that point. Nonetheless, if no degradation of the drug is perceived under these conditions, it should be refluxed in acid/alkali of higher strength and for longer duration of time [29,30,31]. As an alternative, if total degradation is seen after subjecting the drugs to initial condition, acid/alkali strength can be reduced with a decrease in reaction temperature. In fact, temperature and pH are the major factors in stability of the drug/molecule prone to hydrolytic decomposition [29,30,31]. Most of the drugs/molecules’ hydrolysis depend on the relative concentration of hydronium and hydroxyl ions, which allow one to determine the pH at which each drug is optimally stable [29,30,31].

In this work, the diterpene DHA was successfully encapsulated into calcium alginate microspheres and tested in acidic conditions. Alginate microspheres were able to protect DHA against acidic conditions such as those found in the gastric environment, with a PP% around 60.00 ± 0.05%.

##### Photostability Studies

Photostability testing is performed to generate primary degradants (such as UV or fluorescent light) that create conditions to evaluate if a drug substance suffers unacceptable changes under light exposition [31].

In an attempt to prolong the release of pharmaceutical compounds and protect them from atmospheric agents (moisture, light, heat and/or oxidation), study of modern dosage forms as the employment of microencapsulates that incorporate active ingredients within polymers, are a very appropriate and effective process for the purpose sought [26].

DHA was maintained in calcium alginate microspheres over time (for seven days) and against UV in contrast to free (non-encapsulated) DHA. Similarly to previous studies [26], alginate proved to be a very good encapsulant material.

In this study, the DHA-alginate microspheres were very successful in protecting DHA, where the PE% value was high (around 88.12 ± 0.05%)

### 3.4. Toxicological Studies

#### 3.4.1. Sulforhodamine B Assay

One of the most largely used methods for in vitro cytotoxicity screening is the SRB assay [39]. SRB is a strong-intensity bright-pink aminoxanthene dye with two sulfonic groups that bind to basic aminoacidic residues under mild acidic conditions, and dissociate under basic conditions [39]. This assay is based on the stoichiometric linking of SRB to protein components of cells that are fixed to tissue-culture 96-well plates by trichloroacetic acid, being the amount of dye extracted from stained cells directly proportional to the cell mass [39].

The toxicity effect of DHA was tested against the HaCaT cell line (normal human keratinocytes). The effects on growth of HaCaT after continuous treatment of 72 h were assessed by chemosensitivity assay SRB assay. DHA did not reduce significantly the viability number of normal cells, however it showed a half maximum inhibitory concentration (IC_50_) of 19.69 ± 7.40 µg/mL. Nevertheless, DHA was non-toxic to normal cells (HaCaT) in the same range of concentrations in a recent study [40].

#### 3.4.2. Artemia Salina Toxicological Study

*Artemia salina* model is a known, simple, fast and low-cost test and was preliminary used to assess the toxicity of DHA [41]. The compound (LC_50_ above 40%) was further assayed at a concentration of 10 ppm. The number of dead larvae was recorded and used to calculate the median lethal concentration (LC_50_), after 24 h. LC_50_ values less than or equal to 10 ppm were considered active.

A very low mortality of *A. salina* of 21.71 ± 2.18% was observed upon exposure to DHA, demonstrating that the compound in this preliminary toxicological study did not have a toxic effect.

## 4. Discussion

The main goal of this study was the screening of the antimicrobial activity of DHA against a collection of Gram-positive bacteria, Gram-negative bacteria and yeasts, the performance of toxicological assays to assess if the compound had antimicrobial and not cytotoxic properties, and microencapsulation and stability studies to evaluate the chemical behavior and stability of DHA.

In MIC studies, DHA demonstrated to have higher efficacy in Gram-positive bacteria *S. epidermidis* ATCC 12228 and *M. smegmatis* ATCC 607 (both with the lowest MIC value of 7.81 µg/mL) compared with Gram-negative bacteria, where the lowest MIC value of 125 µg/mL was obtained for all clinical isolates of *K. pneumoniae* and clinical isolate strain *E. coli*. The studies of MIC with bioburden have shown that organic challenges have influence in MIC values thus increasing them. In a more recent study [36], some carbazole derivatives from DHA have shown a MIC range between 1.9 and 100 µg/mL against *S. aureus*, but it did not specify which strain. Gu et al. [36] showed that carbazole derivatives from DHA have an enhancement efficacy against Gram-negative bacteria like *E. coli* and also against yeast like *C. albicans*, thus demonstrating the potential of DHA and its derivatives as antimicrobial agents with enhanced antimicrobial activity.

With regard to MIC values with bioburden it was expected that the MIC would have higher concentrations. In a recent study [37], antimicrobial activity with bioburden was tested for some antiseptics, and it was shown that PVP-iodine 10% has an even higher MIC value (1562.5 µg/mL) than DHA against *S. aureus* ATCC 25923. Within the same strain, it was presented that chlorhexidine with bioburden had the same MIC value as DHA, therefore, demonstrating that DHA has interest as an antimicrobial. In traumatic, surgical and burn wound infections, *S. aureus* is the most problematic microorganism. Moreover, there are other such as *P. aeruginosa*, *E. coli* and *K. pneumoniae*, which can cause chronic wound infection [38]. Therefore, it is often necessary the use of systemic and topical antibiotics or the use of antiseptic substances to avoid or eliminate the risk of wound infection. Therefore, it is imperative to find new antimicrobials.

In the biofilm formation assay, most of the tested microorganisms were biofilm producers except for clinical isolate *E. cloacae* HSM 1264. Therefore, this work had a vast collection of microorganisms with biofilm production that will be an advantage for further antimicrobial testing. Regarding the biofilm inhibition properties of DHA, the lowest MBIC value was registered against Gram-positive *S. aureus* ATCC 43866 (0.49 µg/mL; BI 75.13 ± 8.82%), which is a strong biofilm producer. In Gram-positive bacteria with Bioburden challenge only HBS had more influence in MBIC and BI, yet in Gram-negative bacteria all challenges had influence in MBIC and BI values. Notwithstanding, these results demonstrate the capability of this diterpene in the battle against resistant biofilm-producing bacteria. In a recent study [37], MBIC and BI values were obtained for some disinfectants tested against a similar collection of Gram-positive and Gram-negative bacteria and yeasts. Comparatively, DHA show better results than polyhexanide 0.1% (31.25 µg/mL; BI 43.82 ± 0.05%), triclosan 0.15% (46.88 µg/mL; BI 67.90 ± 0.05%) and chlorhexidine 0.2% (62.50 µg/mL; BI 86.14 ± 0.05%) against Gram-positive *S. aureus* ATCC 43866, and similar values against Gram-negative bacteria. MBIC values observed in yeast for the disinfectants was against *C. albicans* ATCC 10231 (62.50 µg/mL; BI 81.83 ± 7.05%).

Pathogenic bacteria can develop mechanisms to adapt and survive in specific microenvironments, which involve overcoming antibiotic killing-effects on them, causing therapeutic failure, recurrence and resistance to the infections. In fact, the biofilms formed by these bacteria existing on the surface of implanted medical devices and in deep tissues, play a meaningful role in persistent infections [39]. This is due to the formation of biofilms that are formed by complex aggregates that involved the bacteria, which are in the dormant state, because of the protection from host defense and antimicrobials by the extracellular matrix where they lived. The extracellular matrix comprising polymeric substances such as DNA, lipopolysaccharides, teichoic acids and proteins. Moreover, biofilm formation and growth, which are accompanied by finely regulated metabolic changes, also affect the bacterial response to antimicrobials [39]. The increase on the emergence of multidrug-resistant bacteria is mainly due to the limited effectiveness against biofilm-related infections [40]. Therefore, it is urgent to investigate new alternatives to these conventional antimicrobial agents, such as DHA and its derivatives.

At concentrations of 31.25, 62.50 and 93.75 µg/mL against the tested organisms (*S. aureus* ATCC 25923), DHA growth curve profiles showed antibacterial properties. This is consistent with the fact that MBC values could not be determined in the concentration range tested against the same microorganisms used to obtain the MIC values, thus suggesting that antimicrobial properties of DHA are mainly due to bacterial growth inhibition.

Regarding microencapsulation and stability studies of DHA, encapsulation efficiency of the compound attained 99.49 ± 0.05% and the stability studies demonstrated that DHA remained stable within alginate microspheres (PP% 60.00 ± 0.05%, PE% 88.12 ± 0.05%). Therefore, this study considers the encapsulation of the diterpene in alginate an excellent way to maintain its physicochemical characteristics.

In the *Artemia salina* toxicological study [41], DHA demonstrated not to have toxicity, and although in the SRB assay it exhibited an IC_50_ value of 19.69 ± 7.40 µg/mL, the 72 h test is considered more stressful towards the HaCaT cells than the 24 h assay as performed by Huh et al. [42]. These preliminary toxicological studies demonstrate that further studies in this area need to be performed using other methods.

To recapitulate, screening of DHA antimicrobial activity evidenced the potential inhibition of planktonic and biofilm growth in a battery of yeasts, Gram-positive and Gram-negative bacteria. Probably the diterpenoid influenced the membrane integrity in all microorganisms and helped to eradicate most biofilm cells. The significant reduction in cell attachment makes diterpenoids an ideal antiadhesive with potential as antibiofilm coating for medical devices. Diterpenes have also been frequently reported to be active against microorganisms [17]. Terpenes are thought to act at different levels in bacteria: (1) disturbing the membrane, thereby, inhibiting respiration and ion transport processes in bacterial cells; (2) altering the lipid composition of the cell membrane, and thus, cell hydrophobicity, which lead to the eradication of the biofilm [18,19]. Soon, it will be imperative to develop and optimize antimicrobial assays, in order to elucidate the mechanisms of action of diterpenes and how they perform to eradicate bacterial biofilms.

The results obtained in this study provide preliminary scientific validation upon the known ethnopharmacological uses of (dehydro)abietic acids. Regardless of the renewed interest on natural compounds and their biological applications, it is decisive to improve the studies of DHA and its derivatives, evaluate their possible uses, understand their safety and unravel their mechanism of action on specific targets.

## 5. Conclusions

One of the most current serious health treats are biofilms infections, mainly because of the existence of antibiotic resistant strains. Using a unique test, as MIC, to assay planktonic (biofilm) susceptibility to antibiotics may be a possible explanation for treatment failure and the appearance of resistances. Though most of the methods performed in this study are time consuming, they are relatively not expensive and easily reproduced. Here, the results of the MIC with or without the Bioburden challenge, MBC, MBIC with or without Bioburden challenge and growth curve study have highlighted the interesting activity of DHA, acknowledge that further studies of this compound are needed: mechanism of action; optimized methods; other studies like minimal biofilm eradication concentration (MBEC), among others; the development of novel compounds with antibacterial and anti-biofilm activity.

## Figures and Tables

**Figure 1 pharmaceutics-13-00825-f001:**
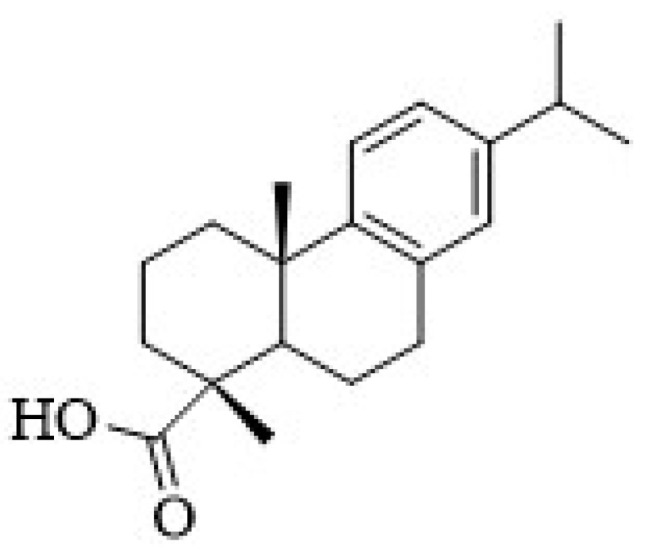
Chemical structure of dehydroabietic acid (DHA).

**Figure 2 pharmaceutics-13-00825-f002:**
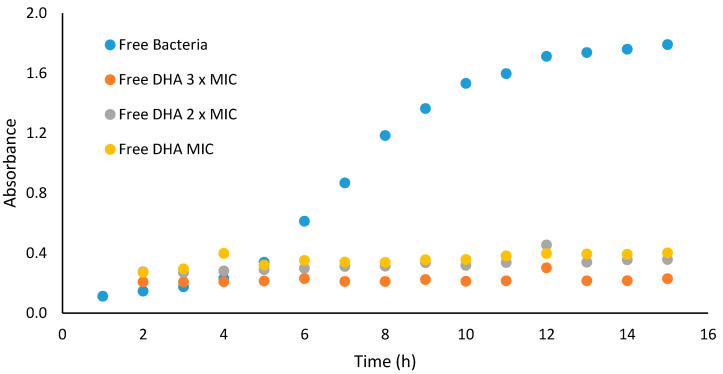
Graphic representation of the growth curve study results of DHA against *S. aureus* ATCC 25923.

**Figure 3 pharmaceutics-13-00825-f003:**
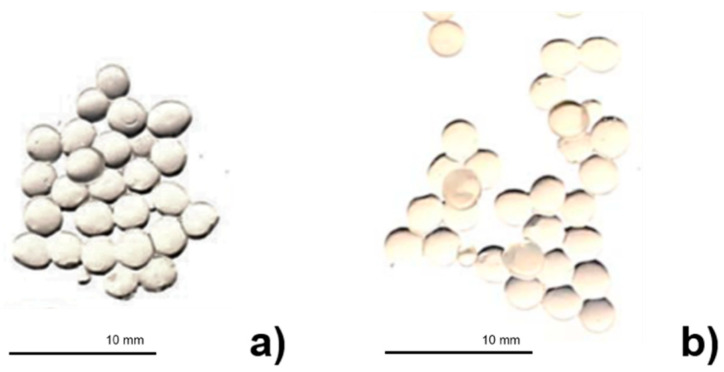
Digital images of (**a**) the empty alginate microspheres and (**b**) the DHA-alginate microspheres.

**Table 1 pharmaceutics-13-00825-t001:** Minimum inhibitory concentration (MIC) values (μg/mL) of DHA against a collection of Gram-positive and Gram-negative bacteria, and some yeasts. Results showed as mean ±SD, *p* < 0.05.

Microorganism	Strain	DHA	Positive Control ^a^	Negative Control ^b^
Gram-positive bacteria	*S. aureus* CIP 106760 ^c^	15.63	RIF > 500	250
	*S. aureus* ATCC 6538	15.63	RIF < 0.49	250
	*S. aureus* ATCC 25923 ^d^	31.25	RIF 0.98	250
	*S. aureus* ATCC 700699	62.5	RIF < 0.49	250
	*S. aureus* FFHB 25923 ^e^	31.25	RIF < 0.49	250
	*S. aureus* ATCC 43300 ^c^	125	RIF < 0.49	250
	*S. aureus* ATCC 43866	>250	RIF < 0.49	250
	*S. epidermidis* ATCC 12228	7.81	RIF < 0.49	250
	*M. smegmatis* ATCC 607	7.81	RIF < 0.49	250
	*E. faecalis* ATCC 29212	>125	VAN 62.5	125
	*E. faecalis* ATCC 51299 ^f^	>125	VAN 62.5	125
	*E. faecalis* V583 ^f^	500	VAN 62.5	500
Gram-negative bacteria	*E. coli* ATCC 25922	>125	AMP < 0.49	125
	*E. coli* HSM 1323 ^c^	>250	AMP < 0.49	250
	*E. coli* HSM 1465 ^c^	>250	AMP < 0.49	250
	*E. coli* HSM 3023 ^c^	125	AMP < 0.49	250
	*P. aeruginosa* ATCC 9027	>125	AMP 31.25	125
	*K. pneumoniae* HSM 701 ^c^	125	AMP < 0.49	250
	*K. pneumoniae* HSM 703 ^c^	125	AMP < 0.49	250
	*K. pneumoniae* HSM 873 ^c^	125	AMP < 0.49	250
	*K. pneumoniae* HSM 2948 ^c^	125	AMP < 0.49	250
	*E. cloacae* HSM 1284 ^c^	>250	AMP > 500	250
Yeasts	*C. albicans* ATCC 10231	>125	NYS 31.25	125
	*S. cerevisiae* ATCC 2601	>62.5	NYS 31.25	62.5

^a^ RIF, rifampicine; VAN, vancomycin; AMP, ampicillin; NYS, nystatin; ^b^ DMSO; ^c^ MRSA; ^d^ MSSA; ^e^ FFHB strain are clinical isolates from Hospital do Barreiro, and HSM strains are clinical isolates from Hospital de Santa Maria deposited at the Microbiology Laboratory, Faculty of Pharmacy, University of Lisbon; ^f^ VRE. All the MIC values were evaluated using DMSO as the solvent, which was also evaluated as the negative control.

**Table 2 pharmaceutics-13-00825-t002:** Minimum inhibitory concentration (MIC) values (μg/mL) of DHA against Gram-positive (*S. aureus* ATCC 25923 and *S. aureus* CIP 106760) and Gram-negative (*P. aeruginosa* ATCC 9027) bacteria. Results showed as mean ± SD, *p* < 0.05.

Bacterial Strain	Biofilm Formation	Free DHA	BSA	HBS
*S. aureus* ATCC 25923 ^a^	+++	31.25	62.5	250
*S. aureus* CIP 106760 ^b^	+	250	500	500
*P. aeruginosa* ATCC 9027 ^c^	+	500	1000	1000

^a^ MSSA; ^b^ MRSA; ^c^ slime producer. All the MIC values were evaluated using DMSO as the solvent, which was also evaluated as the negative control.

**Table 3 pharmaceutics-13-00825-t003:** Minimum biofilm inhibitory concentration (MBIC) values (μg/mL) and percentage of biofilm inhibition for DHA against a collection of Gram-positive and Gram-negative bacteria, and yeast. Results showed as mean ± SD, *p* < 0.05.

Microorganism	Strain		Biofilm Formation ^a^	MBIC (µg/mL)	Biofilm Inhibition (%)
Gram-positive bacteria	*S. aureus* ATCC 25923 ^b^		+++	125	72.66 ± 2.88
	*S. aureus* ATCC 6538		++	500	43.64 ± 1.58
	*S. aureus* ATCC 700699		++	500	57.55 ± 17.50
	*S. aureus* ATCC 43300 ^c^		++	125	65.99 ± 3.72
	*S. aureus* ATCC 43866		+++	0.49	75.13 ± 8.82
	*S. aureus* CIP 106760 ^c^		+	62.5	77.48 ± 0.05
	*S. aureus* FFHB 25923 ^d^		++	3.91	63.69 ± 0.05
	*S. epidermidis* ATCC 12228		++	500	92.56 ± 5.72
	*E. faecalis* ATCC 51299 ^e^		++	7.81	75.93 ± 0.05
	*E. faecalis* ATCC 29212		++	15.63	87.25 ± 0.05
	*E. faecalis* V583 ^e^		++	31.25	75.72 ± 0.05
Gram-negative bacteria	*E. coli* ATCC 25922		+	500	80.22 ± 1.93
	*E. coli* HSM *1323* ^d^		+	250	33.59 ± 0.05
	*E. coli* HSM *1465* ^d^		+	1.95	80.35 ± 13.37
	*E. coli* HSM *3023* ^d^		+++	250	92.58 ± 1.18
	*K. pneumoniae* HSM *701* ^d^		+	0.98	92.75 ± 5.69
	*K. pneumoniae* HSM *703* ^d^		+	0.98	94.13 ± 2.37
	*K. pneumoniae* HSM *873* ^d^		+++	250	98.82 ± 2.00
	*K. pneumoniae* HSM 2948 ^d^		+++	250	96.99 ± 0.24
	*E. cloacae* HSM 1264 ^d^		-	ND	ND
	*P. aeruginosa*	ATCC 9027 ^f^	+	7.81	64.64 ± 10.19
Yeast	*C. albicans*	ATCC 10231	+	62.5	81.83 ± 7.05

^a^ (-) no biofilm producer; (+) weak biofilm producer; (++) moderate biofilm producer; (+++) strong biofilm producer; ^b^ MSSA; ^c^ MRSA; ^d^ FFHB species are clinical isolates from Hospital do Barreiro and HSM strains are clinical isolates from Hospital de Santa Maria, deposited on the Microbiology Laboratory, Faculty of Pharmacy, University of Lisbon; ^e^ VRE; ^f^ slime producer. All the MIC values were evaluated using DMSO as the solvent, which was also evaluated as negative control.

**Table 4 pharmaceutics-13-00825-t004:** Minimum biofilm inhibitory concentration (MBIC) values (μg/mL) and biofilm inhibition (BI) percentage of DHA against Gram-positive (*S. aureus* ATCC 25923 and *S. aureus* CIP 106760) and Gram-negative (*P. aeruginosa* ATCC 9027) with the Bioburden challenge. Results showed as mean ± SD, *p* < 0.05.

Strain	Biofilm Formation	MBIC (µg/mL)			Biofilm Inhibition (%)		
		Free DHA	BSA	HBS	Free DHA	BSA	HBS
*S. aureus* ATCC 25923 ^a^	+++	15.63	15.63	125	62.96 ± 0.15	45.67 ± 0.56	51.54 ± 1.45
*S. aureus* CIP 106760 ^b^	+	250	250	500	50.61 ± 1.56	48.59 ± 4.34	52.03 ± 2.33
*P. aeruginosa* ATCC 9027 ^c^	+	7.81	62.5	31.25	62.15 ± 0.89	43.56 ± 2.12	57.88 ± 1.34

^a^ MSSA; ^b^ MRSA; ^c^ slime producer. All the MBIC values were evaluated using DMSO as the solvent, which was also evaluated as the negative control.

## Data Availability

Data available on request due to restrictions e.g., privacy or ethical. The data presented in this study are available on request from the corresponding author.

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
