# Peer review of "Dehydroabietic Acid Microencapsulation Potential as Biofilm-Mediated Infections Treatment"

_pharmaceutics, 2021, doi:10.3390/pharmaceutics13060825_

Round 1
Reviewer 1 Report
Please see the attached document.

Author Response
RESPONSE TO REVIEWERS’ COMMENTS
Manuscript: 1200600 submitted to Pharmaceutics journal by Iris Neto, Eva María Domínguez-Martín, Epole Ntungwe, Catarina P. Reis, Milica Pesic, Célia Faustino and Patrícia Rijo. This manuscript evaluates the potential of the compound Dehydroabietic Acid encapsulated for treating biofilm-mediated infections.
Dear Editor,
We appreciate the reviewers’ comments, which have helped us to improve the manuscript. We have carefully considered the suggestions, addressing, and incorporating them in the manuscript, using a yellow background to indicate the corrections in the text.
Also, it was included grammatical, orthographical and typographical corrections of English language by the authors throughout the text.
Reviewer #1: Comments to the Authors
The presented manuscript evaluates the antibacterial and anti-biofilm effect of DHA. The authors performed the MIC, MBC, and MBIC assays on most of the 26 selected bacteria; bioburden challenges on 3 strains; and time-kill kinetic study on S. aureus ATCC25923. Though a lot of results are presented, some shortcomings in the manuscript need to be addressed:
Comment 1: Section 2.4 Bacterial strains: S. aureus is not a VRE. The authors put three S. aureus strains as VRE. Also, I believe ATCC 51299 is a E. faecalis, not S. aureus. In addition, the authors should explain why culturing them in anaerobically while most strains are aerobes.
Authors: The reviewer is correct, sorry for that, we changed all these errors from the manuscript. Of course, S. aureus is not a VRE and CIP 106760 and S. aureus FFHB 25923 were changed to MRSA. In addition ATCC 51299 is a E. faecalis which was corrected and, of course we cultured them aerobically.
Comment 2: Section 2.7.4 Biofilm formation assay: Crystal violet assay has limitations and concerns as it stains the bacteria and their extracellular matrix so the biofilm viability cannot be determined. The authors should explore additional assay for biofilm quantification. I believe the group of no biofilm producer (OD595 > ODc) was defined incorrectly. In addition, it seems that the authors denote ODc for both cut-off OD value and OD of the negative control. The authors should clarify the 4 groups and the equation (1).
Authors: The reviewer is correct, the Crystal violet assays has its limitations and concerns, nonetheless, in this case the staining was done only after washing the plates and fixating the biofilms, which reduces the staining of residual bacteria. Washing of biofilm is a step of the utmost importance since it is supposed to remove all non-adherent cells while simultaneously providing preservation of the biofilm integrity. The biofilm integrity is check throughout the entire process since bacterial growth until biofilm fixation measuring the turbidity in the wells. We used the Crystal violet staining because it was the most frequently used stain cited and its use is acceptable because the former washing steps wash off all non-adherent cells, so only the resting adherent cells will be stained.
The four groups and the equation where also corrected and clarified:
The cut-off OD value (ODc) was defined as three standard deviations (SD) above the mean OD of the negative control (equation 1). The OD value of a tested strain was expressed as its average. The strains were divided into the following groups: OD595 ≤ ODc = no biofilm producer (-); ODc < OD595 ≤ 2 x ODc = weak biofilm producer (+); 2 ´ ODc < OD595 ≤ 4 x ODc = moderate biofilm producer (++); and 4 ´ ODc < OD595 = strong biofilm producer (+++).
Comment 3: Section 2.7.7 Time-kill kinetics studies: Why this assay is only performed for one strain? The authors determined the CFU but plotted the absorbance against time on Figure 2. Absorbance cannot be used to show the number of viable cells (Line 461).
Authors: We thank the reviewer for the comment. The S. aureus ATCC 25923 was the only strain used because it was a high biofilm producer and we have chosen this strain to use as a model because it is well documented bacteria. Further tests will be performed after optimizing the process of ACA encapsulation.
Comment 4: Table 3: The data of S. aureus ATCC 700699, 51299, M. smegmatis ATCC 607, and S. cerevisiae ATCC 25922 is missing. The data of S. aureus ATCC 700699 appear on Table 3.
Authors: We thank for the reviewer comment and the data from S. aureus ATCC 700699 is not missing from table 3, we have checked and is the fourth bacteria mentioned. S.aureus ATCC 51299 is instead the bacterial strain E. faecalis ATCC 51299 and the values unfortunately were duplicated in these two columns, and everything is rectified. M. smegmatis ATCC 607 due to loss of sample was not selected for biofilm testing. S. cerevisiae ATCC 25922 is actually ATCC 2601 and is used for negative control, we apologize for the errors and they were all corrected.
Comment 5: Section 3.4.2: Authors assayed the compound at concentration of 10 ppm and claimed that it does not have toxicity effect. 10 ppm is 10 μg/mL, which is lower/much lower than most of the MIC, MBC, and MBIC. With the SRB assay exhibited an IC50 of 19.69 ± 7.40 μg/mL, I think it is premature for the author’s claim about the toxicity of DHA.
Authors: The reviewer is correct, for Artemia salina which is just a screening assay the maximum concentration to test is 10 ppm, so we complemented with cytotoxicity using SRB assay and thus more assays should be performed about toxicity.
Comment 6: Minor points:
- Line 34: 700,000: The number “700.000” has been changed to “700,000”.
- Line 36: remove “a” : The “a” has been removed.
- Lines 62-63: Rephase needed: The sentence has been rephrased to “The abietanes possess a characteristic aromatic ring C, being an example, the dehydroabietic acid”.
- Section 2.10: One-way or two-way ANOVA? One-way ANOVA was mostly used.
- Table 1. cloacae instead of Ent. Cloacae: The correction has been introduced in the text and also in Table 3.
- Please use the same format for the strain column for both Tables 1 and 3: Tables 1 and 3. were modified to appear in the same format.
- Table 3 and Line 383: cloacae 1284? Yes, it was corrected in the indicated lines and also in line 588.
- Figure 2: No curve and SD were presented on this figure. Yes, is data not shown because is not relevant.
- Line 484: 0.17 mm? : It was corrected to 37±0.20 and 2.31± 017 X 103 µm
Authors: All minor corrections were performed and added to the manuscript.
Finally, the authors should check the spelling, grammar, and sentences in the manuscript with an appropriate native language service. Many paragraphs only contain one sentence. Formatting is an issue as well.
Authors: English spelling, grammar and format has been checked and corrected.
Thank you so much for all the comments,
Kind Regards

Reviewer 2 Report
The article by Neto et al. evaluates the antimicrobial activity of dehydroabietic acid (DHA) against a collection of sensitive and resistant bacteria and yeasts through the minimum inhibitory concentration (MIC), MIC with bio-burden challenge, minimum bactericidal concentration (MBC), minimum biofilm inhibitory concentration (MBIC), MBIC with bio-burden challenge, and time-kill kinetic studies. Toxicological assays were carried out to evaluate if DHA had antimicrobial and not cytotoxic properties, and finally, microencapsulation and stability studies were performed to evaluate the chemical behavior and stability of DHA. The research design and the methods were adequate and the data collection obtained is extensive and of interest to the field. The manuscript is recommended for publication but minor corrections should be performed in the text:
- In my opinion, several sentences included in the Results should be transferred to the Discussion because of the comparison of the observed antimicrobial activity of DHA with other antimicrobial compounds described in the literature. For example, lines 327-328: In a more recent study [34], some carbazole derivatives from DHA have shown a MIC range between 1.9–100 µg/mL against S. aureus, but it did not specify which strain.
- Please review the text because there are a few grammar and English typos.
- Please improve the quality of the tables.
Author Response
RESPONSE TO REVIEWERS’ COMMENTS
Manuscript: 1200600 submitted to Pharmaceutics journal by Iris Neto, Eva María Domínguez-Martín, Epole Ntungwe, Catarina P. Reis, Milica Pesic, Célia Faustino and Patrícia Rijo. This manuscript evaluates the potential of the compound Dehydroabietic Acid encapsulated for treating biofilm-mediated infections.
Dear Editor,
We appreciate the reviewers’ comments, which have helped us to improve the manuscript. We have carefully considered the suggestions, addressing, and incorporating them in the manuscript, using a green background to indicate the corrections in the text.
The article by Neto et al. evaluates the antimicrobial activity of dehydroabietic acid (DHA) against a collection of sensitive and resistant bacteria and yeasts through the minimum inhibitory concentration (MIC), MIC with bio-burden challenge, minimum bactericidal concentration (MBC), minimum biofilm inhibitory concentration (MBIC), MBIC with bio-burden challenge, and time-kill kinetic studies. Toxicological assays were carried out to evaluate if DHA had antimicrobial and not cytotoxic properties, and finally, microencapsulation and stability studies were performed to evaluate the chemical behavior and stability of DHA. The research design and the methods were adequate and the data collection obtained is extensive and of interest to the field. The manuscript is recommended for publication but minor corrections should be performed in the text:
Comment 1:
In my opinion, several sentences included in the Results should be transferred to the Discussion because of the comparison of the observed antimicrobial activity of DHA with other antimicrobial compounds described in the literature. For example, lines 327-328: In a more recent study [34], some carbazole derivatives from DHA have shown a MIC range between 1.9–100 µg/mL against S. aureus, but it did not specify which strain.
Authors: We thank for your comment. We took your opinion in consideration an altered the several sentences that you purposed.
Comment 2:
Please review the text because there are a few grammar and English typos.
Authors: We thank for your comment. English spelling, grammar and format has been checked and corrected.
Comment 3:
Please improve the quality of the tables.
Authors: We thank for your comment. All tables were improved.

Reviewer 3 Report
Brief Summary
The manuscript pharmaceutics-1200600 reports results on the screening of antimicrobial and antibiofilm activity of dehydroabietic acid (DHA) against sensitive and resistant Gram-positive and Gram-negative bacteria and yeasts. The DHA was further microencapsulated and stability studies were carried out to evaluate its chemical behaviour and stability.
Broad comments
- Abstract: Authors should describe the main results obtained before the statement in L23-25.
- Introduction: The introduction correctly places the study in a broad context and highlight why it is important. The authors clearly defined the purpose of the work and its significance, as well as the specific hypotheses being tested.
- Materials and Methods: Except for Bio-burden challenge, the authors described with sufficient detail the methods used. I suggest authors add details about all companies mentioned and describe better the Bio-burden challenge for the readers not involved in pharmaceutical microbiology.
- Results: The results description is clear and precise. However, the authors did not show in the Tables the statistical significance. Authors should provide it in all the test where it was performed. Moreover, this section should only contain the description of the results, avoiding the discussion of the same.
- Discussion: Authors discussed the results from the perspective of previous studies within the results section. I suggest moving these sentence in the discussion section and creating a clear flow of concepts. In this section, the authors should avoid the repetition of numeric results and discuss the findings from the perspective of previous studies and of the working hypotheses in the broadest context possible. Thus, this section should be strongly improved.
- Conclusions: The section is appropriate as the study contains many elements. However, it is again only a repetition of the results. In this section, the authors should highlight the strength and the limitations of the work and present future research directions.
Specific comments
L95 – The authors should provide more details about DHA obtained from this company.
L60-61 – The authors should provide citations of the previous work published.
L293 – Add a version of GraphPad Prism as well as the companies details. Do the same also for the others.
The English language should be revised in the entire manuscript.
Author Response
RESPONSE TO REVIEWERS’ COMMENTS
Manuscript: 1200600 submitted to Pharmaceutics journal by Iris Neto, Eva María Domínguez-Martín, Epole Ntungwe, Catarina P. Reis, Milica Pesic, Célia Faustino and Patrícia Rijo. This manuscript evaluates the potential of the compound Dehydroabietic Acid encapsulated for treating biofilm-mediated infections.
Dear Editor,
We appreciate the reviewers’ comments, which have helped us to improve the manuscript. We have carefully considered the suggestions, addressing, and incorporating them in the manuscript, using a teal background to indicate the corrections in the text.
Also, it was included grammatical, orthographical and typographical corrections of English language by the authors throughout the text.
Reviewer #3: Comments to the Authors
The manuscript pharmaceutics-1200600 reports results on the screening of antimicrobial and antibiofilm activity of dehydroabietic acid (DHA) against sensitive and resistant Gram-positive and Gram-negative bacteria and yeasts. The DHA was further microencapsulated and stability studies were carried out to evaluate its chemical behaviour and stability.
Broad comments:
Comment 1:
- Abstract: Authors should describe the main results obtained before the statement in L23-25.
Authors: We thank you for your comments. The results will be added in the abstract.
Comment 2:
- Introduction: The introduction correctly places the study in a broad context and highlight why it is important. The authors clearly defined the purpose of the work and its significance, as well as the specific hypotheses being tested.
Authors: We thank for your comment.
Comment 3:
- Materials and Methods: Except for Bio-burden challenge, the authors described with sufficient detail the methods used. I suggest authors add details about all companies mentioned and describe better the Bio-burden challenge for the readers not involved in pharmaceutical microbiology.
Authors: We thank for your comment. We have corrected and added what you purposed.
Comment 4:
- Results: The results description is clear and precise. However, the authors did not show in the Tables the statistical significance. Authors should provide it in all the test where it was performed. Moreover, this section should only contain the description of the results, avoiding the discussion of the same.
Authors: We thank for your comment. We will add to the tables the statistical significance and improve the Results section.
Comment 5:
- Discussion: Authors discussed the results from the perspective of previous studies within the results section. I suggest moving these sentence in the discussion section and creating a clear flow of concepts. In this section, the authors should avoid the repetition of numeric results and discuss the findings from the perspective of previous studies and of the working hypotheses in the broadest context possible. Thus, this section should be strongly improved.
Authors: We thank for your comment. We will strongly improve this section.
Comment 6:
- Conclusions: The section is appropriate as the study contains many elements. However, it is again only a repetition of the results. In this section, the authors should highlight the strength and the limitations of the work and present future research directions.
Authors: We thank for your comment. We will highlight what you have purposed.
Comment 7:
Specific comments
L95 – The authors should provide more details about DHA obtained from this company. Manufacturer :
FUJIFILM Wako Pure Chemical Corporation
Storage Condition :
Keep at RT.
CAS RN® :
1740-19-8
Molecular Formula :
C20H28O2
Molecular Weight :
300.44
L60-61 – The authors should provide citations of the previous work published.
L293 – Add a version of GraphPad Prism as well as the companies details. Do the same also for the others.
The English language should be revised in the entire manuscript.
Authors: We thank for your comment. We will add everything that you have purposed. English spelling, grammar and format has been checked and corrected.

Reviewer 4 Report
The authors did t studies regarding DHA as an anti-microbial and anti-biofilm agent. But there are major questions,
1. However, the author needs to explain the mechanism in more detail.
2. It is difficult to understand how microencapsulated beads cause the anti-biofilm effect.
3. As the EPS of the biofilms very strong what caused DHA to disperse that gave anti-microbial activity, that is the main challenge for researchers.
4. Line no. 616, the authors completely ignored to discuss how this compound bypass EPS of biofilm.
5. it is not clear how CV staining is carried out.
Author Response
RESPONSE TO REVIEWERS’ COMMENTS
Manuscript: 1200600 submitted to Pharmaceutics journal by Iris Neto, Eva María Domínguez-Martín, Epole Ntungwe, Catarina P. Reis, Milica Pesic, Célia Faustino and Patrícia Rijo. This manuscript evaluates the potential of the compound Dehydroabietic Acid encapsulated for treating biofilm-mediated infections.
Dear Editor,
We appreciate the reviewers’ comments, which have helped us to improve the manuscript. We have carefully considered the suggestions, addressing, and incorporating them in the manuscript, using a pink background to indicate the corrections in the text.
Also, it was included grammatical, orthographical and typographical corrections of English language by the authors throughout the text.
Reviewer #4: Comments to the Authors
Comments:
The authors did t studies regarding DHA as an anti-microbial and anti-biofilm agent. But there are major questions,
- However, the author needs to explain the mechanism in more detail.
Authors: We thank you for the comment. Unfortunately, the mechanisms of action of DHA (diterpenes) is not entirely understood. Terpenes are thought to act at different levels in bacteria: 1) disturbing the membrane, thereby, inhibiting respiration and ion transport processes in bacterial cells; and 2) altering the lipid composition of the cell membrane, and thus, cell hydrophobicity which lead to the eradication of the biofilm. Considering this, it will be imperative to develop and optimize antimicrobial assays, in order to elucidate the mechanisms of action of diterpenes and how they perform to eradicate bacterial biofilms
- It is difficult to understand how microencapsulated beads cause the anti-biofilm effect.
Authors: We thank you for your comment. DHA-alginate beads were not tested for the anti-biofilm effect in this work, only free DHA was tested. Further testing with micro- and nanoencapsulation will be performed in a near future and studies about the mechanism of action of DHA on bacteria and biofilms must me enhanced and performed.
- As the EPS of the biofilms very strong what caused DHA to disperse that gave anti-microbial activity, that is the main challenge for researchers.
Authors: We thank you for your comment. As we mentioned before more studies must be conducted to investigate which mechanism of action is taken with free and encapsulated DHA.
- Line no. 616, the authors completely ignored to discuss how this compound bypass EPS of biofilm.
Authors: We thank you for your comment. As we mentioned before more studies must be conducted to investigate on how this compound bypasses EPS of biofilm.
- it is not clear how CV staining is carried out.
Authors: We thank you for your comment. We include and resume how the method was done, however it is an extensive protocol and reference procedure.

Reviewer 5 Report
The manuscript pharmaceutics-1200600 reports the study of the antimicrobial and antibiofilm activity of dehydroabietic acid (DHA) against a consistent number of gram positive and gram negative bacteria as well as yeasts. Alginate spheres entrapped with DHA were also prepared and characterized in terms of stability (at low pH and at UV irradiation) and cytotoxicity.
Major comment
The authors performed a significant amount of experimental work with different tests and a remarkable number of bacterial species. However, I have a serious concern on an experimental aspect, to which the authors never referred to, which is DHA water solubility. Usually, acidic resins have a very low water solubility. For instance, abietic acid is considered water insoluble. From a quick literature search, I have found an old study by Nyren V and Back E (Acta Chimica Scandinavica 12 (1958) 1516-1520) in which the solubility of DHA is compared with that of Abietic acid. Actually, DHA is more soluble than abietic acid because of the presence of the aromatic ring instead of the conjugated double bond. However, DHA water solubility is reported to be 6.6 mg/L which is a concentration extremely lower than those used by the authors in the experiments. For instance, at page 4 line 138, DHA MIC was explored in a 50,000-0.49 mg/L range. The same is for the other performed antimicrobial tests. When DHA was entrapped in alginate, a concentration of even 10 mg/mL (line 225) was used. Therefore, to have their manuscript published, the authors need to explain such incongruence and eventually report in the manuscript the DHA solubility concentration.
Minor comments
- Line 116. Please explain why all microrganisms were incubated in an anaerobic work station.
- Line 135. Please add the antibiotic used as positive controls.
- Line 235. Briefly, add some information on the UV method used for determination of DHA concentration (e.g. the wavelength and the DHA concentration linearity range)
- Lines 358-363. Please explain why BSA and HBS were selected for the bio-burden experiments
- Figure 3. Which type of microscope was used to take the images. Add the information if M&M.
- Line 497. The authors stated that DHA is an hydrophilic compound. I wouldn’t define it so.
Author Response
RESPONSE TO REVIEWERS’ COMMENTS
Manuscript: 1200600 submitted to Pharmaceutics journal by Iris Neto, Eva María Domínguez-Martín, Epole Ntungwe, Catarina P. Reis, Milica Pesic, Célia Faustino and Patrícia Rijo. This manuscript evaluates the potential of the compound Dehydroabietic Acid encapsulated for treating biofilm-mediated infections.
Dear Editor,
We appreciate the reviewers’ comments, which have helped us to improve the manuscript. We have carefully considered the suggestions, addressing, and incorporating them in the manuscript, using a blue background to indicate the corrections in the text.
Also, it was included grammatical, orthographical and typographical corrections of English language by the authors throughout the text.
Reviewer #5: Comments to the Authors
The manuscript pharmaceutics-1200600 reports the study of the antimicrobial and antibiofilm activity of dehydroabietic acid (DHA) against a consistent number of gram positive and gram negative bacteria as well as yeasts. Alginate spheres entrapped with DHA were also prepared and characterized in terms of stability (at low pH and at UV irradiation) and cytotoxicity.
Major comment
Comment 1:
The authors performed a significant amount of experimental work with different tests and a remarkable number of bacterial species. However, I have a serious concern on an experimental aspect, to which the authors never referred to, which is DHA water solubility. Usually, acidic resins have a very low water solubility. For instance, abietic acid is considered water insoluble. From a quick literature search, I have found an old study by Nyren V and Back E (Acta Chimica Scandinavica 12 (1958) 1516-1520) in which the solubility of DHA is compared with that of Abietic acid. Actually, DHA is more soluble than abietic acid because of the presence of the aromatic ring instead of the conjugated double bond. However, DHA water solubility is reported to be 6.6 mg/L which is a concentration extremely lower than those used by the authors in the experiments. For instance, at page 4 line 138, DHA MIC was explored in a 50,000-0.49 mg/L range. The same is for the other performed antimicrobial tests. When DHA was entrapped in alginate, a concentration of even 10 mg/mL (line 225) was used. Therefore, to have their manuscript published, the authors need to explain such incongruence and eventually report in the manuscript the DHA solubility concentration.
Authors: We thank the reviewer for the comment and the reviewer is correct. All DHA samples were diluted in DMSO.
Minor comments
- Line 116. Please explain why all microrganisms were incubated in an anaerobic work station.
We apologize, the microorganisms were incubated aerobically.
- Line 135. Please add the antibiotic used as positive controls.
Antibiotics positive controls were added.
- Line 235. Briefly, add some information on the UV method used for determination of DHA concentration (e.g. the wavelength 620 nm, and the DHA concentration linearity range).
All purposed data will be added e.g. the wavelength 620 nm etc)
- Lines 358-363. Please explain why BSA and HBS were selected for the bio-burden experiments.
BSA and HBS were used in the bioburden experiments because they can mimic wound environment, medical devices contamination or other type of contamination. Bioburden testing is an integral part of microbiological monitoring programs when investigating antimicrobial or disinfectant compounds, and they are also employed during manufacturing processes to optimize sterilization protocols.
- Figure 3. Which type of microscope was used to take the images. Add the information if M&M.
No microscope was needed to photograph de DHA-alginate beads due to its size. A Xiaomi Redmi 4, 5-element lens, 1.12μm pixels, 13 MP CMOS camara was utilized, which will be added to the materials section.
- Line 497. The authors stated that DHA is an hydrophilic compound. I wouldn’t define it so.
In Line 497 it is mentioned that DHA and alginate have the same hydrophilic properties, being them low hydrophilic compounds, does not state that DHA is hydrophilic. We will add “low” for better understanding.
Authors: We thank the reviewer for the comments that help us to improve the manuscript. All the corrections were performed accordingly.

Round 2
Reviewer 1 Report
I thank the authors for their efforts to rectify and improve this version of the manuscript. There is only one remaining concern regarding figure 2 that authors haven’t addressed. Once the authors address the concerns on figure 2, the manuscript can be accepted and published.
Both live and dead cells contribute to absorbance. Therefore, the authors cannot claim “…reduction in number of viable cells…” (Line 441).
Author Response
We thank the reviewer comments and we agree to change the following sentence:
“ The time-kill kinetics profile of DHA against the test microorganisms at test concentrations showed a reduction in number of cells over the first four hours when compared to the control (microorganism without antimicrobial agent).”

Reviewer 3 Report
I revised the new version of manuscript no. pharmaceutics-1200600. My previous comments have been carefully addressed. The only additional minor change that I suggest is to provide in the Tables the case letters of the Tukey's test that you performed.
Author Response
Thank you for your kind comment. In this project no Tukey test was needed to p value significance. All triplicates tested demonstrated p < 0.05. English language was carefully checked and confirmed after this comment.

Reviewer 4 Report
The authors defended all the comments, I recommend the acceptance of the paper.
Author Response
We thank the reviewer comments that help us to improve the manuscript.

Reviewer 5 Report
I find the manuscript is written in a very superficial way and also serious flaws are present. I would have preferred a narrower selection of strains but a greater accuracy in performing the experiments and in writing the manuscript.
The fact that the authors have omitted that DHA was solubilized in DMSO represents a serious carelessness.
Still it is not clear which is the negative control. At lines 146, 147 the authors stated that negative control was sterile culture media. I wonder why they didn’t test DMSO at used concentrations as negative control or if they did it why they don’t write it clearly.
For what I understand, DHA was solubilized in pure DMSO and then 100 microliters were added in the media broth (100 microliter). That means, DMSO was 50% that is a very high concentration. DMSO is suggested to be kept at 10% or less for biological assays.
All of the DHA MIC have to be considered in DMSO? Which is the significance of such results?
In Table 1 and 2, the authors stated “Results showed as mean±SD, p<0.05”, but I don’t see any SD in the tables. In these tables, how was negative control prepared?
In table 3. Minimum biofilm inhibitory concentration (MBIC), the negative control is not reported.
Finally, to my comment “The authors stated that DHA is an hydrophilic compound. I wouldn’t define it so” the authors replied: “in Line 497 it is mentioned that DHA and alginate have the same hydrophilic properties, being them low hydrophilic compounds, does not state that DHA is hydrophilic. We will add “low” for better understanding”. This is incorrect. Alginate is a very hydrophilic compound. In contrast, DHA is hydrophobic.
Author Response
We appreciate the reviewers’ comments, which have helped us to improve the manuscript. We have carefully considered the suggestions, addressing, and incorporating them in the manuscript, using a yellow background to indicate the corrections in the text.
Also, it was included grammatical, orthographical and typographical corrections of English language by the authors throughout the text.
Reviewer #5: Comments to the Authors
Comment 1: I find the manuscript is written in a very superficial way and also serious flaws are present. I would have preferred a narrower selection of strains but a greater accuracy in performing the experiments and in writing the manuscript.
Authors: Thanks for the sincere comment, that we agreed. However, as the present manuscript is a preliminary screening study, it shows a general view of the microbial strains, sensitive and resistant to the compound. In the future, it is planned to design a new study focusing on a narrower selection of strains as the reviewer suggests.
Comment 2: The fact that the authors have omitted that DHA was solubilized in DMSO represents a serious carelessness.
Authors: Thanks very much for the comment, we want to apologize for the error. DMSO was the negative control in all tests and assays, so this information was added to the manuscript.
Comment 3: Still it is not clear which is the negative control. At lines 146, 147 the authors stated that negative control was sterile culture media. I wonder why they didn’t test DMSO at used concentrations as negative control or if they did it why they don’t write it clearly.
Authors: Thanks very much for your kind comment. DMSO was diluted in sterile culture media as the solvent of the compounds so it was used as negative control like other previous results from other projects (1, 2, 3, 4). Previous tests confirmed that DMSO has not relevant antimicrobial action (around 125-250 µg/mL) but we read and applied their results considering these negative results. Therefore, we used sterile culture media like previous studies as a control for bacteria grow (1, 2, 3, 4) so we used different controls for compounds solvent and also bacteria grow.
- Matias D, Nicolai M, Fernandes AS, Saraiva N, Almeida J, Saraiva L, Faustino C, Díaz-Lanza AM, Reis CP, Rijo P. Comparison Study of Different Extracts of Plectranthus madagascariensis, P. neochilus and the Rare P. porcatus (Lamiaceae): Chemical Characterization, Antioxidant, Antimicrobial and Cytotoxic Activities. Biomolecules. 2019 May 8;9(5):179. doi: 10.3390/biom9050179. PMID: 31072074; PMCID: PMC6571840.
- Ntungwe E, Domínguez-Martín EM, Teodósio C, Teixidó-Trujillo S, Armas Capote N, Saraiva L, Díaz-Lanza AM, Duarte N, Rijo P. Preliminary Biological Activity Screening of Plectranthus spp. Extracts for the Search of Anticancer Lead Molecules. Pharmaceuticals (Basel). 2021 Apr 23;14(5):402. doi: 10.3390/ph14050402. PMID: 33922685.
- Neto Í, Andrade J, Fernandes AS, Pinto Reis C, Salunke JK, Priimagi A, Candeias NR, Rijo P. Multicomponent Petasis-borono Mannich Preparation of Alkylaminophenols and Antimicrobial Activity Studies. ChemMedChem. 2016 Sep 20;11(18):2015-23. doi: 10.1002/cmdc.201600244. Epub 2016 Jul 26. PMID: 27457409.
- Rimpiläinen T, Nunes A, Calado R, Fernandes AS, Andrade J, Ntungwe E, Spengler G, Szemerédi N, Rodrigues J, Gomes JP, Rijo P, Candeias NR. Increased antibacterial properties of indoline-derived phenolic Mannich bases. Eur J Med Chem. 2021 Apr 20;220:113459. doi: 10.1016/j.ejmech.2021.113459. Epub ahead of print. PMID: 33915373.
Comment 4: For what I understand, DHA was solubilized in pure DMSO and then 100 microliters were added in the media broth (100 microliter). That means, DMSO was 50% that is a very high concentration. DMSO is suggested to be kept at 10% or less for biological assays.
Authors: Thanks very much for the comment. The reviewer is correct. In biological assays with cells, DMSO is suggested to be kept at 10% or less, however in bacterial assays (1,2,3,4,5), DMSO can be used up to 80% without any alterations on bacterial growth and we confirmed that and is usually described on previous works which is usually around 125-250 µg/mL (1,2,3,4,5).
1- Brito RC de, Silva GN da, Farias TC, Ferreira PB, Ferreira SB. Standardization of the Safety Level of the Use of DMSO in Viability Assays in Bacterial Cells. MDPI sciforum. 2017;3.
2- Matias D, Nicolai M, Fernandes AS, Saraiva N, Almeida J, Saraiva L, Faustino C, Díaz-Lanza AM, Reis CP, Rijo P. Comparison Study of Different Extracts of Plectranthus madagascariensis, P. neochilus and the Rare P. porcatus (Lamiaceae): Chemical Characterization, Antioxidant, Antimicrobial and Cytotoxic Activities. Biomolecules. 2019 May 8;9(5):179. doi: 10.3390/biom9050179. PMID: 31072074; PMCID: PMC6571840.
3- Ntungwe E, Domínguez-Martín EM, Teodósio C, Teixidó-Trujillo S, Armas Capote N, Saraiva L, Díaz-Lanza AM, Duarte N, Rijo P. Preliminary Biological Activity Screening of Plectranthus spp. Extracts for the Search of Anticancer Lead Molecules. Pharmaceuticals (Basel). 2021 Apr 23;14(5):402. doi: 10.3390/ph14050402. PMID: 33922685.
4- Neto Í, Andrade J, Fernandes AS, Pinto Reis C, Salunke JK, Priimagi A, Candeias NR, Rijo P. Multicomponent Petasis-borono Mannich Preparation of Alkylaminophenols and Antimicrobial Activity Studies. ChemMedChem. 2016 Sep 20;11(18):2015-23. doi: 10.1002/cmdc.201600244. Epub 2016 Jul 26. PMID: 27457409.
5- Rimpiläinen T, Nunes A, Calado R, Fernandes AS, Andrade J, Ntungwe E, Spengler G, Szemerédi N, Rodrigues J, Gomes JP, Rijo P, Candeias NR. Increased antibacterial properties of indoline-derived phenolic Mannich bases. Eur J Med Chem. 2021 Apr 20;220:113459. doi: 10.1016/j.ejmech.2021.113459. Epub ahead of print. PMID: 33915373.
Comment 5: All of the DHA MIC have to be considered in DMSO? Which is the significance of such results?
Authors: Thanks very much for the comment. Yes, all must be considered in DMSO because is the solvent that solubilized the DHA, which is not soluble in water or medium. The DMSO influence in bacterial growth was studied as negative control, therefore the significance of the results demonstrates that DHA has antimicrobial properties, and that DHA needs to be explored as an antibacterial compound and that further testing is required.
Comment 6: In Table 1 and 2, the authors stated, “Results showed as mean ± SD, p<0.05”, but I don’t see any SD in the tables. In these tables, how was negative control prepared?
Authors: Thanks very much for the comment. The SD is not showed in some of the tables, particularly in MICs, due to SD being null (triplicate). Other reviewers asked to put “mean±SD, p<0.05” in all tables which was included.
The negative control (DMSO) was prepared in the same way as the positive controls or sample: “Using a multichannel micropipette, a serial dilution was made to 1:2 proportion between each row of wells (50,000–0.49 mg/L range).”
Comment 7: In table 3. Minimum biofilm inhibitory concentration (MBIC), the negative control is not reported.
Authors: Thanks very much for the comment. This appears in Lines 176-177. “Sterile culture media was used as a negative control for bacteria.”
Comment 8: Finally, to my comment “The authors stated that DHA is an hydrophilic compound. I wouldn’t define it so” the authors replied: “in Line 497 it is mentioned that DHA and alginate have the same hydrophilic properties, being them low hydrophilic compounds, does not state that DHA is hydrophilic. We will add “low” for better understanding”. This is incorrect. Alginate is a very hydrophilic compound. In contrast, DHA is hydrophobic.
Authors: Thanks very much for the comment. It was a misleading. We are very sorry. Alginate is a hydrophilic polymer, having several guluronic and mannuronic monomers which link several compounds like DHA. DHA has carboxylic group that might interact with alginate. Structurally different from the others resins like isopimaric acid, DHA has three double bonds whilst the others have only two, but it is described that DHA has water solubility around 5.1 mg/mL. This low solubility is not a limiting factor to interact with hydrophilic alginate. Even being weak interaction, it is our aim to have DHA in a free form and act as anti-biofilm agent.

Round 3
Reviewer 5 Report
1) At lines 146, 147, the authors have to describe better the negative control, adding the information about DMSO as negative control and relevant bibliography stating DMSO is not toxic for bacterial cells at so high concentrations.
2) This sentence at lines 474-476 needs to be deleted. "This great value of EE% was foreseen, as alginate and DHA have the same low hydrophilic properties". Again, DHA and alginate have not the same low hydrophilic properties.
3) In the captions of all of the tables, add that the MIC values are in DMSO (%).
Author Response
Reviewer #5: Comments to the Authors
Comment 1: 1) At lines 146, 147, the authors have to describe better the negative control, adding the information about DMSO as negative control and relevant bibliography stating DMSO is not toxic for bacterial cells at so high concentrations.
Authors: Thanks for the sincere comment, that we agreed, and we add the following sentence:
“Additionally, DMSO was used as negative control as the solvent used to dissolve DHA, at this high concentration DMSO was not toxic as previously described [15-17, 27].”
Comment 2: 2) This sentence at lines 474-476 needs to be deleted. "This great value of EE% was foreseen, as alginate and DHA have the same low hydrophilic properties". Again, DHA and alginate have not the same low hydrophilic properties.
Authors: Thanks very much for the comment, we want to apologize for the error. The sentence was deleted.
Comment 3: In the captions of all the tables, add that the MIC values are in DMSO (%).
Authors: We agree with the reviewer and in all the caption tables we add: “all the MIC values were evaluated using DMSO as the solvent which was also evaluated as negative control”. We did not add the percentage because is different in all the wells due to the 1:2 successively dilutions and thus variable in all the concentrations tested. This sentence was added on MIC and MBIC values on the corresponding Tables.
